# Time of Day Influences Concentrations of Total Protein and Albumin in Cerebrospinal Fluid in HIV

**DOI:** 10.3390/ijms24032832

**Published:** 2023-02-01

**Authors:** Visesha Kakarla, Scott L. Letendre, Ronald J. Ellis

**Affiliations:** HIV Neurobehavioral Research Center (HNRC), University of California San Diego, San Diego, CA 92093, USA

**Keywords:** HIV, cerebrospinal fluid, glymphatic, biomarker

## Abstract

The accumulation of soluble proteins and metabolites during wakefulness and their clearance during sleep via the glymphatic system occurs in healthy adults and is disturbed in some neurological conditions. Such diurnal variations in the cerebrospinal fluid (CSF) proteins produced outside the central nervous system and entering via the blood–brain barrier (BBB) have not been evaluated in people with HIV (PWH). CSF and blood were collected in 165 PWH at six US centers between 2003 and 2007. The time of collection was compared to CSF albumin, globulin, and total protein concentrations using bivariate and multivariate regression. Participants all took antiretroviral therapy (ART) and were mostly middle-aged (median age 44.0 years) men (78.8%), with AIDS (77.0%), plasma HIV RNA ≤ 200 copies/mL (75.8%), and immune recovery (median CD4+ T-cell count 414/µL). CSF was collected at median 1:10 p.m. (range 9:00 a.m. to 5:20 p.m.) and within a median of 15 min of blood collection. A later time of CSF collection was associated with higher total protein (*p* = 0.0077) and albumin (*p* = 0.057) in CSF but not in serum, and was additionally associated with higher CSF globulin (*p* = 0.013). The glymphatic clearance of albumin, globulin, and total protein is preserved in PWH. The analyses of soluble biomarkers in CSF should account for the time of collection.

## 1. Introduction

The glymphatic system is important in the clearance of proteins, metabolites, waste products, and toxins from the central nervous system (CNS). This system operates by promoting the removal of these molecules from the CNS extracellular fluid (ECF) into the lymphatics [1]. For example, albumin, which lacks specific transporters to cross the blood–brain barrier (BBB), exits via perivascular and periaxonal/perineural routes [2]. This clearance ultimately reduces the concentrations of waste products in the cerebrospinal fluid (CSF), which receives important contributions from the ECF as well as from choroid plexus production [3]. Glymphatic clearance occurs primarily during sleep, leading to a diurnal pattern in which CSF levels of these molecules fall during sleep when the glymphatic system is more active, resulting in lower levels early in the morning, and rise over the course of the day when the glymphatic system is less active, resulting in higher levels in the afternoon and evening [4]. If this system is not properly functioning, the normal diurnal pattern is disrupted [5]. Conditions that disrupt this variation include sleep disturbances, aquaporin-4 mutations, and neurodegenerative disorders such as Alzheimer’s disease [6].

Additionally, the dysfunction of the BBB, seen in several immune-altering conditions, may allow for entry of other cell types and proteins and thus subsequent removal via the glymphatic system [7]. For example, while the brain exhibits immune privilege, BBB disruption may permit entry of B cells into or prevent clearance of B cells from the brain parenchyma, allowing for locally produced immunoglobulins to proliferate and exert local CNS damage requiring glymphatic clearance [8]. Given that total protein is composed primarily of albumins and globulins (specifically immunoglobulins), both may ultimately be affected by alterations to the glymphatic system [9]. In addition, it is thought that disruption of the glymphatic system in the setting of these immune-altering conditions may result in impaired clearance from the CNS of inflammatory cytokines [10].

The diurnal variation of albumin, globulins, and other blood-derived molecules found in the CSF such as inflammatory cytokines has not been studied in people with HIV (PWH). Clearance of these proteins might be influenced by the glymphatic system. We evaluated whether diurnal variations occur in CSF albumin, globulin, and total protein. We predicted that, if glymphatic clearance is at least partially preserved in PWH, then CSF levels of these components would be lower in the morning, following glymphatic clearance during sleep, as compared to later in the day.

## 2. Results

### 2.1. Participants

Participants were 165 PWH who enrolled in the CHARTER study between 2003 and 2007 who had CSF and matched plasma stored. The median age of the participants was 44.0 years (IQR 39.0, 48.0), and 130 (78.8%) identified as male. Black participants were 42.4% of the cohort; 13.9% participants were Hispanic and 43.6% were non-Hispanic white. Median (IQR) nadir CD4+ T-cell count was 101/µL (20, 210) and median (IQR) CD4+ T-cell count was 414/µL (259, 533) at the time of evaluation. Median estimated duration of HIV infection was 11.1 years (IQR 6.7, 15.6) and 127 (77.0%) had a diagnosis of AIDS. All participants had been taking antiretroviral therapy (ART) for at least 1 month at the time of data collection, with the median duration of the current ART regimen being 8.7 months (IQR 4.1, 17.1). Most participants had HIV RNA ≤ 200 copies/mL in plasma (125, 75.8%) and in CSF (148, 89.7%).

### 2.2. Relationship of Collection Time to Albumin, Globulin, and Total Protein

The median CSF total protein was 41 mg/dL (IQR 32, 49); 12% of values were above the upper limit of the reference range (60 mg/dL). Higher CSF total protein correlated with a later CSF collection time (r = 0.207, *p* = 0.0077; Figure 1a), and this relationship was similar for the subgroup of participants with suppressed plasma HIV RNA (*n* = 123, r = 0.167, *p* = 0.0644). The relationship between serum total protein and CSF collection time was weaker and non-significant (r = 0.146, *p* = 0.0618). The median (IQR) CSF albumin was 16 (13, 22) mg/dL; 1.8% were above the upper limit of the reference range for males 18–50 years (10–45 mg/dL). A similar relationship was seen for CSF albumin: the later collection time more strongly correlated with higher albumin in CSF (r = 0.148, *p* = 0.057, Figure 1b) than in serum (r = 0.112, *p* = 0.153). Additionally, the later CSF collection time significantly correlated with higher globulin in CSF (r = 0.195, *p* = 0.013, Figure 1c).

The median (IQR) CSF-to-serum albumin ratio (CSAR) was 3.95 (3.16, 5.13); 8.6% of ratios were above the reference range for persons under 45 years (6.8). The CSF collection time did not correlate with the CSAR (r = 0.139, *p* = 0.075). 

Threshold values in the data were explored with recursive partitioning, which identified that mid-afternoon may be important for CSF collection. Participants who had CSF collected after 2:50 pm (35, 21.2%) had higher concentrations of albumin (mean 21.8 vs. 17.4 mg/dL, Cohen’s d = 0.488, Figure 2a) and total protein (mean 50.0 vs. 40.6 mg/dL, Cohen’s d = 0.523, Figure 2b) in CSF. Neither albumin nor total protein in serum differed compared to this CSF collection time, although recursive partitioning indicated that they did differ by a blood collection time of 11:30 a.m. (serum albumin: *p* = 0.0127, serum total protein: *p* = 0.00765).

### 2.3. Relationship of Collection Time to Viral and Cellular Biomarkers

Among participants with detectable HIV RNA in CSF, the CSF collection time did not correlate with HIV RNA in CSF (r = 0.178, *p* = 0.406). Similarly, the blood collection time did not correlate with HIV RNA in plasma among participants with detectable levels (r = −0.092, *p* = 0.471). Fifteen participants (9.09%) had CSF pleocytosis (white cells > 5/µL). CSF collection time did not correlate with CSF glucose (r = 0.0401, *p* = 0.616), or with CSF leukocyte (r = −0.0460, *p* = 0.560) or erythrocyte (r = −0.00546, *p* = 0.945) counts. In addition, IL-6 (r = 0.9899, *p* = 0.792), IP-10 (r = −0.884, *p* = 0.331), and MCP-1 (r = 0.00653, *p* = 0.966) were not associated with the time of CSF collection.

### 2.4. Accounting for Possible Confounding Conditions

Other variables were considered potential confounders of the relationship between CSF collection time and the outcomes of interest (CSF albumin, total protein and globulins) when these variables were significantly related to both CSF collection time and the outcome. CSF collection time was not related to gender (*p* = 0.832), ethnicity (*p* = 0.436), current CD4 (*p* = 0.946), nadir CD4 (*p* = 0.999), or viral detectability in blood (*p* = 0.288) or CSF (*p* = 0.916). Older participants tended to undergo lumbar punctures later in the day (r = 0.192; *p* = 0.0135); however, age was not correlated with CSF levels of total protein (r = 0.0577, *p* = 0.467), albumin (r = 0.123, *p* = 0.116), IL-6 (r = 0.0414, *p* = 0.811), IP-10 (r = −0.138, *p* = 0.404) or MCP-1 (r = −0.0345, *p*= 0.824).

## 3. Discussion

We found that CSF albumin and total protein concentrations (but not other biomarkers including CSF leukocytes, glucose and inflammatory cytokines) were higher in participants whose lumbar punctures (LPs) were performed later in the day compared to earlier. The effect size for this relationship (Cohen’s d) was medium when comparing those whose LPs were performed later in the afternoon to those carried out earlier in the day. This finding was not explained by changes in blood–CSF barrier function (the CSF/serum albumin ratio) or by differences in serum albumin concentration, and was robust in consideration of the demographic and other potential confounders. We propose that diurnal variation in glymphatic clearance from CSF accounts for these results. 

The major source of CSF albumin is its transport from the blood, via binding glycoprotein receptors on epithelial cells in the choroid plexus, and subsequent transfer into the ventricular CSF [3]. CSF is renewed approximately four times per day [11]. The clearance of albumin in the CSF is relatively fast; in the mouse, when albumin was administered into the CSF, only ~6% remained in the CSF after 1 h [12]. Globulins are also largely produced outside of the central nervous system, with intrathecal production being primarily limited to the production of immunoglobulins by intracranial malignancy or aberrant immune processes [13].

The glymphatic system is responsible for clearing soluble proteins and other molecules from the central nervous system by shunting these molecules, accumulated during the day in the interstitial fluid and CSF, into the lymphatic system. In people without HIV (PWoH), this results in a diurnal pattern in which levels of macromolecules, such as albumin and globulins, increase over the course of the day as they accumulate in the CSF and decrease at night as they are cleared through the glymphatic system. Until now, it was unknown whether this pattern exists in PWH, who are commonly prone to sleep disturbances which could interfere with glymphatic clearance, as occurs in other neurodegenerative conditions, such as Alzheimer’s disease (AD) [14,15].

Alternative explanations for our findings may be proposed. For example, diurnal changes in albumin/globulin/protein entry, CSF production, blood–brain and blood–CSF barrier permeability, or transport via trafficking cells might occur that could in turn affect albumin, globulin, and total protein. Arguing against some of these hypotheses, however, we did not observe a relationship between time of day and CSF cell counts or CSAR. 

Recognition of the diurnal pattern is important in the context of CSF collection for clinical or research studies, as timing of collection may be a confounding factor. Therefore, time should be recorded and considered when collecting levels of CSF components, such as albumin.

The strengths of this study are the relatively large number of PWH who underwent lumbar puncture and the racial and ethnic diversity of the cohort, enhancing generalizability. We replicated a similar diurnal pattern across several CSF components of relevance to HIV. Several limitations may be noted. CSF was collected only during daytime hours; thus, further data would need to be collected to determine if albumin levels decrease over the course of sleep. We did not collect data on sleep quantity or quality to determine if these might influence the findings. Female participants were underrepresented relative to males, although the proportion (21%) was comparable to the percent of women with HIV in the U.S. (20–25%). The rate of viral suppression was lower than in many modern cohorts; however, the correlation between CSF protein and CSF collection time was similar for those with and without viral suppression. 

Future directions might include performing studies involving matched PWoH to assess the extent to which diurnal variations are preserved in PWH. Future studies also might determine the impact of various sleep parameters on CSF albumin, globulin, total protein and HIV RNA. The CSF molecules of importance for CNS function, such as metabolic waste products cleared by the glymphatic system, would be of particular interest. 

## 4. Materials and Methods

### 4.1. Design and Participants

We performed a cross-sectional analysis of people with HIV (PWH) from the CNS HIV AntiRetroviral Therapy Effects Research (CHARTER) study, which enrolled research volunteers at six U.S. sites: Johns Hopkins University, Icahn School of Medicine at Mount Sinai, University of Texas Medical Branch Galveston, University of California San Diego, University of Washington, and Washington University in St. Louis. Participants were all ambulatory, community-dwelling PWH. Inclusion criteria were willingness to undergo lumbar puncture (LP) and phlebotomy. Exclusions were active opportunistic disease; inherited or acquired bleeding disorder or other contraindications to LP; active, uncontrolled psychotic disorder; or neurological disorder unrelated to HIV, such as multiple sclerosis or Parkinson’s disease. The protocol was approved by Institutional Review Board at each research site. All participants provided informed consent for the study procedures outlined by the protocol. Visits described here were conducted between 2003 and 2007.

### 4.2. Clinical and Lab Procedures

CSF was collected by lumbar puncture performed by trained clinicians. Current antiretroviral therapy (ART) was gathered through a structured interview. Albumin, total protein, and HIV RNA were measured in CSF and blood, which were collected within 90 min of each other (median 15 min, interquartile range [IQR]: 25 min before to 25 min after blood), using standard methods. Unless noted, most samples presented here were measured at local, CLIA-certified reference labs. Serum albumin was measured as part of the comprehensive metabolic panel using a standard colorimetric assay. Serum globulin was calculated by subtracting albumin from total protein. CSF albumin was measured by immunonephelometry (ARUP Laboratories, Salt Lake City, UT). Expected albumin ranges in healthy adults were 0–35 mg/dL (CSF) and 3.5–5.2 g/dL (serum). CSF total protein was measured by spectrophotometry. We calculated the CSF-to-serum albumin ratio as an index of blood–brain barrier permeability. Peripheral blood CD4+ T cell count was measured by flow cytometry. CSF cells were counted by automated methods. HIV RNA was quantified in plasma and CSF by reverse transcriptase-polymerase chain reaction (Roche Ultrasensitive HIV-1 Monitor, Branchburg, NJ) with a lower limit of quantification of 50 copies/mL. In a subset of participants, CSF levels of cytokines were also available, measured using commercially available immunoassay kits manufactured by Millipore (Burlington, MA): interleukin-6 (IL-6; *n* = 36), interferon gamma-induced protein 10 (IP-10; aka CXCL10; *n* = 39) and monocyte chemoattractant protein 1 (MCP-1; *n* = 44)

### 4.3. Statistical Analysis

Demographic and clinical characteristics were summarized using numbers and percentages, means and standard deviations, and medians and IQR as appropriate. Albumin and HIV RNA concentrations were compared to the time of collection of either CSF or blood using bivariate and multivariate regression. Recursive partitioning was used to identify potentially informative threshold values in the time of day. We calculated the CSF-to-serum albumin ratio (CSAR) as an index of blood–CSF barrier (BCB) permeability. Potential confounders assessed included demographics, current and nadir CD4+ T-cells, viral suppression status (plasma HIV RNA copies ≤ or >200/mL) and CSF leukocyte counts. When potential confounders such as age, demographic and disease variables were significantly related to both the predictor (CSF collection time) and outcomes of interest (e.g., CSF protein), we evaluated these further in multivariable regression analyses. We calculated effect sizes as Cohen’s d. Analyses were conducted using JMP Pro version 15.0.0 (SAS Institute Inc., Cary, NC, USA, 2018). 

## Figures and Tables

**Figure 1 ijms-24-02832-f001:**
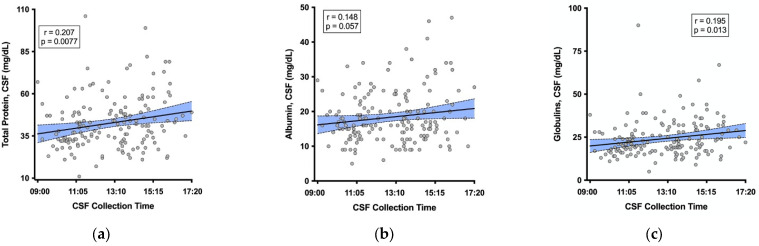
Later CSF collection time correlated with higher (**a**) CSF total protein, (**b**) CSF albumin, and (**c**) CSF globulin. Shaded area represents the 95% confidence band around the fit regression line.

**Figure 2 ijms-24-02832-f002:**
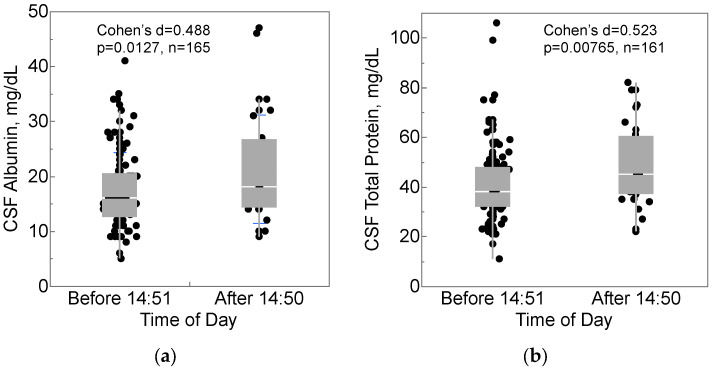
CSF collection time after 2:50 pm was associated with higher (**a**) CSF albumin and (**b**) CSF total protein. Box-and-whisker plots show medians (middle horizontal line), interquartile ranges (box), ranges (whiskers), and Cohen’s d, effect size.

## Data Availability

Not applicable.

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
