# Peer review of "Time of Day Influences Concentrations of Total Protein and Albumin in Cerebrospinal Fluid in HIV"

_ijms, 2023, doi:10.3390/ijms24032832_

Round 1
Reviewer 1 Report
This study is focused on determining the variations in the amount of albumin and total protein due to the time of day in which the sample is taken. It is a very interesting study that provides relevant information, since it has implications for establishing the best time to take samples for the study of various biomarkers in different pathologies of the central nervous system. However, several points need to be considered before considering this manuscript for publication.
1. The title must include the type of patients in which the study was developed
2. Abbreviations are used without these being previously described throughout the text
3. The footer of image 2 does not correspond to this one.
4. Figure 2 should show the individual points that correspond to each boxplot
5. All images submitted must include the n, and the statistics used in each case.
6. In the methodology, it is important that each of the determinations be described in detail. If kits have been used, these must specify which brand was used in each case.
7. In line 184, it is not understood what determination was made with immunonephelometry.
8. On line 185, it says “expected range” of what? Albumin or globulins?
9. Why was cytokine detection only done in some study subjects and not all? How did you consider this for confounding variables?
10. The authors must attach a table in the results section showing the blood and serum levels of each of the determinations that were made.
11. In lines 85-89 the figures are misquoted
Author Response
- Included “in HIV” to represent the patient population that was studied.
- Abbreviations have now been defined and corrected throughout the document.
- Box-and-whisker plots now show medians (middle horizontal line), interquartile ranges (box) and ranges (whiskers).
- Graphs have been reconstituted and updated as requested.
- N values have been updated.
- Additions were made to the methodology section.
- Corrected line 184 to include that CSF albumin was determined using immunonephelometry.
- Corrected line 185 to include to specify that the normal ranges were provided for CSF and serum albumin.
- Cytokines were available on a sample of convenience comprising participants whose samples were used for other research projects unrelated to this this one. The number of participants with cytokine measurements was small, limiting these use of these data in multivariable analyses.
- Thank you for your suggestion – would you please clarify what metrics you would like provided in the table?
- Misquoted figures have been edited.
Reviewer 2 Report
I do not require any changes or additions.
The glymphatic system is a proposed pathway for the clearance of proteins and macromolecules from the brain, and disruption of glymphatic flow is implicated in neurological diseases.
Total clearance of a substance is the sum of clearance values for all available pathways, including the perivascular pathways and the blood-brain barrier.
The novelty of this research is to have analysed the glymphatic system by studying the relationship between collection time and albumin, globulin and total protein in the cerebrospinal fluid of HIV-positive patients over a period of time
Author Response
Thank you for your review.
Round 2
Reviewer 1 Report
The authors took into account the comments of the reviewers. The manuscript is clearer now, they just need to add the information on the kits used for the determination of cytokines by flow cytometry (lines 200-202)
Author Response
Cytokine kit information has now been included in the manuscript.